# Using multi-label ensemble CNN classifiers to mitigate labelling inconsistencies in patch-level Gleason grading

**Muhammad Asim Butt[1]\*, Muhammad Farhat Kaleem[2], Muhammad Bilal [3,4], Muhammad Shehzad Hanif[3,4]**

**1** Department of Electrical Engineering, University of Management and Technology, Lahore, Pakistan,
**2** School of Engineering, University of Management and Technology, Lahore, Pakistan, **3** Center of Excellence in Intelligent Engineering Systems, King Abdulaziz University, Jeddah, Saudi Arabia,
**4** Department of Electrical and Computer Engineering, King Abdulaziz University, Jeddah, Saudi Arabia

\* asim.butt@umt.edu.pk

**Data Availability Statement:** Publicly available datasets were analyzed in this study. This data can be found here: https://data.mendeley.com/datasets/9xxm58dvs3/2 The results presented in this paper can be reproduced through the source

## Abstract

This paper presents a novel approach to enhance the accuracy of patch-level Gleason grading in prostate histopathology images, a critical task in the diagnosis and prognosis of prostate cancer. This study shows that the Gleason grading accuracy can be improved by addressing the prevalent issue of label inconsistencies in the SICAPv2 prostate dataset, which employs a majority voting scheme for patch-level labels. We propose a multi-label ensemble deep-learning classifier that effectively mitigates these inconsistencies and yields more accurate results than the state-of-the-art works. Specifically, our approach leverages the strengths of three different one-vs-all deep learning models in an ensemble to learn diverse features from the histopathology images to individually indicate the presence of one or more Gleason grades (G3, G4, and G5) in each patch. These deep learning models have been trained using transfer learning to fine-tune a variant of the ResNet18 CNN classifier chosen after an extensive ablation study. Experimental results demonstrate that our multi-label ensemble classifier significantly outperforms traditional single-label classifiers reported in the literature by at least 14% and 4% on accuracy and f1-score metrics respectively. These results underscore the potential of our proposed machine learning approach to improve the accuracy and consistency of prostate cancer grading.

## Introduction

Prostate cancer is one of the most common types of cancer in men, posing significant challenges in diagnosis and treatment. Traditional diagnostic methods, such as biopsy followed by histopathological examination, are invasive and subject to inter-observer variability. With the advent of digital pathology, the potential for computer-aided diagnosis has opened up, promising more accurate and consistent results. In this regard, various researchers have considered Deep learning, a subset of machine learning, which has shown remarkable success in image recognition tasks, making it a promising tool for digital pathology as well. In recent years,

codes found here: https://github.com/
MuhammadAsimButt/sicap_multi_label.

**Funding:** This research work was funded by
Institutional Fund Projects under grant no. (IFPIP:
1825-135-1443). The authors gratefully
acknowledge technical and financial support
provided by the Ministry of Education and
Deanship of Scientific Research (DSR) at King
Abdulaziz University, Jeddah, Saudi Arabia. The
funders had no role in study design, data collection
and analysis, decision to publish, or preparation of
the manuscript.

**Competing interests:** The authors have declared
that no competing interests exist.

there has been a surge of research exploring the application of deep learning methodologies to digital pathology in prostate cancer. These studies have spanned a range of tasks, from pre-processing tasks like quality assessment and staining normalization, to diagnostic tasks like cancer detection and Gleason grading, and even prediction tasks such as recurrence prediction or genomic correlations [1]. The research in this area has been fueled by the fact that conventional image recognition tasks and the analysis of whole slide images (WSIs) in digital pathology share several similarities, which make deep learning techniques highly applicable and beneficial for both. Moreover, transfer learning, a powerful technique in deep learning, where a model trained on one task is repurposed on a related task has been instrumental in this field since in digital pathology, annotated data can be scarce [2]. Thus, various well-known neural network architectures pre-trained for general purpose image recognition tasks have been readily adapted by the researchers for digital pathology domain. Network fine-tuning as well as using activations from inner layers as features have been tried. Thus, by leveraging pre-trained models, researchers can overcome the challenge of limited annotated data in the field of digital pathology and improve the performance of deep learning models in detecting and classifying prostate cancer from WSIs [3–6]. However, as suggested by Rabilloud et al. [7] and Abut et al. [8], there is still room for improvement and more work is needed to validate these models externally and ensure their robustness in real-world clinical settings. It is particularly important to note here that while there are similarities with the general-purpose imagery, there are also unique challenges in digital pathology, such as the need for extremely high-resolution image analysis, that require specialized adaptations of these techniques.

Ruiz-Fresneda et al. [9] have provided a study which examines worldwide scientific output on the application of machine learning to the most significant types of cancer, using a range of bibliometric measures. On similar lines, Morozov et al. [10] have provided a comprehensive review of the precision of various Artificial Intelligence (AI) techniques in diagnosing and grading prostate cancer based on histological analysis. Their conclusion was that the precision of AI in identifying and grading Prostate Cancer (PCa) matches that of skilled pathologists. This promising method has numerous potential clinical uses, leading to faster and more efficient pathology reports. However, they also cautioned that the implementation of AI in routine practice may be hindered by the complex and time-consuming process of training and fine-tuning convolutional neural networks. Akinnuwesi et al. [11] have explored the utility of a conventional machine learning algorithm i.e. Support Vector Machine (SVM) on a small dataset [12]. They have reported 98.6% accuracy for the binary classification task. Other researchers such as Li et al. [13] have considered deep learning approaches in prostate cancer diagnosis using Magnetic Resonance Imaging (MRI). However, while MRI is more accurate than WSI testing, it still faces several challenges such as increased cost, lack of broad availability, differences in MRI acquisition and interpretation protocols. Moreover, WSI is particularly useful for Gleason grading and has been considered widely by the researchers. For instance, Mandal et al. [14] have investigated transfer learning for adapting well-known CNN architectures to the task of cancer detection. However, WSIs require careful processing as noted by Kanwal et al. [15] and Foucart et al. [16]. Another recent study has developed a deep learning model that uses gigapixel pathology images and slide-level labels for prostate cancer detection and Gleason grading [17]. The model first crops whole-slide images into small patches and extracts features from these patches using a deep learning model trained with self-supervised learning. Tabatabaei et al. [18, 19] have considered the problem of making manual annotation less laborious through automated retrieval of similar cancerous patches using a CNN-based autoencoder. Recently, Morales-Álvarez et al. [20] have proposed the use of multiple instance learning to tackle the problem of patch-level label generation by exploiting the correlation among neighboring patches. Although 95.11% accuracy has been reported on SICAPv2 dataset, this

study is limited to the binary cancer detection task and Gleason grading has not been considered.

A critical component of prostate cancer diagnosis and prognosis is 'Gleason Grading', a system used to evaluate the stage of prostate cancer using prostate biopsy samples. However, it presents several challenges. There is often considerable inter-observer variability even among expert pathologists, which could lead to unnecessary treatment or missing a severe diagnosis. This makes the task of Gleason grading difficult and subjective due to the need for visual assessment of cell differentiation and Gleason pattern predominance. In a bid to come up with a robust automated method for Gleason scoring through deep learning, researchers have commonly employed patch-based detection. This method involves dividing the whole slide images (WSIs) into smaller, manageable 'patches' of images, which are then analyzed individually [21, 22]. An initial study by Speier et al. [22] proposed an automatic patch selection process based on image features. This algorithm segments the biopsy and aligns patches based on the tissue contour to maximize the amount of contextual information in each patch. The patches are then used to train a fully convolutional network (machine learning model) to segment high grade, low grade, and benign tissue from a set of histopathological slides. Another similar study used a convolutional neural network (CNN) for automated detection of Gleason patterns and determination of the grade groups [23]. The outcome of the CNN was subsequently converted into probability maps, and the grade group of the whole biopsy was obtained according to these probability maps. In a similar approach proposed by Schmidt et al. [24], a multi-class grading with an F1-score of 0.72 has been reported. However, both works do not reproduce the detailed results on the four validation and one test sets of SICAPv2 dataset. The latter work has also reported an F1-score of 0.81 on the PANDA dataset which is another comprehensive collection of prostate cancer biopsies used for the Prostate Cancer Grade Assessment (PANDA) Challenge [25]. This dataset consists of almost 11,000 biopsies available as whole-slide images of hematoxylin and eosin (H&E) stained tissue specimens. Similar to SICAPv2, the grading process for this dataset also involves finding and classifying cancer tissue into Gleason patterns (3, 4, or 5) based on the architectural growth patterns of the tumor. Pati et al. [26] have considered both segmentation of the WSIs as well as the classification at the patch-level on three different datasets including SICAP. However, their reported F1 score on the latter is merely 0.65 which is lower than the previously reported results in the literature. Golfe et al. [27, 28] have taken another innovative approach to improve the Gleason grading in the wake of insufficient, imbalanced and poorly labelled training examples. Specifically, they have trained a generative network to artificially create more training examples than are available in the original SICAP dataset. The main idea is to enhance the classification accuracy through more variations in the training data. They have reported an average accuracy and F1-score of 0.71 and 0.67 respectively which is a marginal improvement over the work of Silva-Rodríguez et al. [29].

Ambrosini et al. [30] have trained a custom CNN for the detection of cribriform pattern which is a specific arrangement of cells that is seen in WSIs for some types of cancer, including prostate cancer. It is characterized by small, round or oval glands that are arranged in a sieve-like pattern. The cribriform pattern is thought to be associated with more aggressive cancers, and it may be a factor in determining a patient's prognosis. They have reported a mean area under the curve of up to 0.81 in sensitivity vs false positives graph. In comparison, Silva-Rodríguez et al. [29] have also considered detection of cribriform pattern with a score of 0.82. However, the comparison is inconclusive since very different datasets have been employed by these two studies.

Another notable recent effort is the introduction of SICAPv2 dataset [29] consisting of 155 biopsies (WSIs) from 95 different patients. The dataset has pixel-level Gleason Grading (GG) labeled through consensus of expert pathologists. The authors claimed unprecedented

detection accuracy using a custom CNN architecture to classify the GG labels at the patch level. However, as noted earlier, the labelling of WSIs being a tedious task, this dataset also suffers from inexact and incomplete pixel-level labelling [31]. Specifically, the patch-level labelling has been conventionally done through majority vote of how each pixel in the patch is labeled according to the Gleason grade. This approach, however, leads to at least three problems i.e.

**Loss of Information:** The majority voting scheme potentially ignores the information related to the minority classes which are inevitably present in numerous patches.

**Misclassification:** If the patch contains a mix of different Gleason grades, the majority voting scheme could result in misclassification since it is highly sensitive to the manual pixel-level labelling done by the pathologists.

**Labelling Noise:** The presence of label noise can negatively affect the training performance of the machine learning models which rely heavily on the accuracy of the provided labels at the patch level.

The labelling noise in histopathology datasets has been recognized by researchers and various techniques have been suggested to mitigate the effects of the same. For instance, Karimi et al. [32] have suggested model regularization to alleviate this problem. Similarly, Ashraf et al. [33] have proposed a unique loss-based method for denoising patch-based labels for Gastric cancer histopathology images. However, the problem of labelling noise due to the presence of more than one type of Gleason Grades has not been addressed. To this end, the work described in this paper,

- Provides a statistical analysis of the patch-level labelling noise in SICAPv2 prostate histology dataset.

- Proposes an ensemble machine learning classifier to detect all occurrences (multi-labels) of Gleason grades at the patch level, rather than just the majority grade.

- Provides an open-source framework for Gleason grading at patch and WSI-level based on the proposed ensemble classifier to facilitate researchers and practitioners working in the field of digital histopathology.

## Materials and methods

This section describes our approach to accurately assign Gleason grades to the patches extracted from WSIs for prostate cancer using a multi-label approach. Our methodology leverages the power of CNNs and the concept of transfer learning for recognizing intricate imaging patterns for classification. Specifically, we utilize pre-trained CNN architectures as the backbone of our model, capitalizing on their proven ability to extract robust features from image data. The models are trained and validated using the SICAPv2 dataset, a comprehensive collection of prostate histopathology images with annotated Gleason grades. This approach allows us to harness the existing knowledge encapsulated in these architectures and adapt it to the specific task of Gleason grade detection. Moreover, the multi-label approach enables the model to predict multiple Gleason grades that may be present in a single image, thereby providing a more nuanced understanding of the disease's severity. Specific details of the proposed detection framework have been given in the following sub-sections. The effectiveness of the proposed approach has been validated by comparing it with state-of-the-art results reported in the literature using various metrics, as detailed in Section 4.

### Dataset overview

The focus of this research is the SICAPv2 dataset, comprising 155 biopsies from 95 individual patients. WSIs have been obtained from tissue samples by slicing, staining and ultimately

digitizing. Skilled urogenital pathologists reviewed these slides and assigned a unified Gleason score to each biopsy. The distribution of primary Gleason grades (GG) in the biopsies is as follows: 36 noncancerous regions, 40 samples with Gleason grade 3 (GG3), 64 with Gleason grade 4 (GG4), and 15 with Gleason grade 5 (GG5). To handle the large WSIs, they were downsampled to 10x resolution and segmented into 512x512 patches with a 50% overlap. A tissue presence mask for the patches was generated using the Otsu threshold method. Patches with less than 20% tissue were excluded for model development aimed at predicting the main Gleason grade. The database comprises 4417 non-cancerous patches, 1635 labelled as GG3, 3622 as GG4, and 665 as GG5. It's important to note that in cases where a patch contained multiple annotated grades, the label of the predominant grade was assigned. Additionally, 763 GG4 patches also contain annotated cribriform glandular regions. To facilitate model training and optimize the involved hyperparameters, the dataset has been partitioned by the original authors using a cross-validation approach. Specifically, each patient was exclusively allocated to one-fold to prevent overestimation of system performance and ensure its generalization. As such, the database was split into 5 groups (i.e. Val1, Val2, Val3, Val4 and Test), each containing roughly 20% of the patches. It's important to highlight that this division aimed to maintain class balance across sets.

## Statistical insight into patch-level labelling inconsistencies

As previously outlined, Gleason grading for prostate cancer involves two distinct labelling approaches at the patch-level and whole slide image (WSI) level, each with its own set of benefits and challenges. The patch-level labelling method assigns labels to individual "patches" within a WSI, allowing for a detailed tissue analysis. This is particularly beneficial when a single WSI contains multiple Gleason grades. However, this method requires significant time and expertise for manual annotation of each patch. As mentioned before, SICAPv2 dataset has provided patch-level labels to facilitate classification. This scheme requires that if a patch contains more than one annotated grade, the label typically assigned is the majority grade. This practice can lead to several issues. For instance, it may result in information loss about other grades present within the same patch, potentially oversimplifying the tissue's complexity and heterogeneity. Additionally, the majority grade may not accurately represent the entire patch's characteristics. For example, a patch might contain a substantial amount of a higher Gleason grade, but if it's not the majority, it could be overlooked, potentially underestimating the disease's severity. There can also be variability in the assignment of the majority grade among different observers, leading to label inconsistencies. This is especially true in cases where the distribution of different grades within a patch is nearly equal. Lastly, a model trained on such data might not perform well in real-world scenarios where multiple Gleason grades are present in a single patch. Thus, a patch might contain a substantial amount of a higher Gleason grade, but if it's not the majority, it could be overlooked, potentially underestimating the disease's severity. To address these issues, this work has proposed multi-label classification which allows each patch to be assigned multiple labels corresponding to the different Gleason grades present, accurately capturing the tissue's complexity and heterogeneity.

To appreciate the level of label inconsistencies in SICAPv2 dataset, several statistics have been collected in this study. Fig 1 shows three example patches where pixels depict a variety of different grades (mask manually annotated by expert histopathologists) while the label has been assigned based on simple majority vote. Thus, a significant number of pixels in a patch could be misclassified due to a higher level of granularity. Fig 2 provides a statistical insight into the label inconsistency problem by showing probability distributions of misclassified pixel belonging to different grades/classes in the patches belonging to SICAPv2 dataset partition

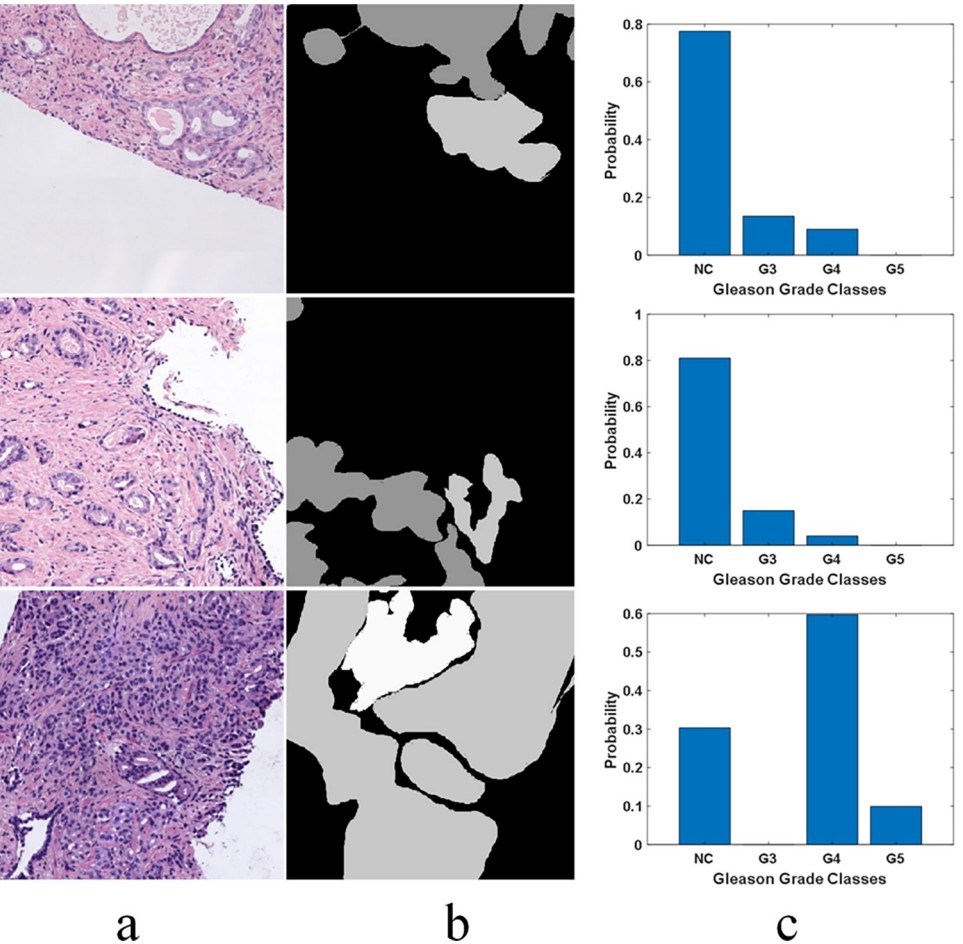

**Fig 1. Examples of misclassified pixels due to majority voting-based labelling in SICAPv2 dataset.** The pixels belonging to the minority class don't get acknowledged separately (a) RGB Patch; (b) Labelling Mask; (c) Probability distribution estimate of the grades/classes represented in the Mask.

'Val1'. It can be observed that upto 30% of pixels could be mislabeled if they don't belong to the majority class.

The misclassification statistics for SICAPv2 dataset have been summarized in Table 1. It can be noticed that while on average only up to 2% of pixels are misclassified in a given set, as many as 30% could be misclassified in individual instances. This misclassification can potentially lead to suboptimal performance while training a machine learning classifier especially in the wake of high level of imbalance. In the light of these observations, this study suggests implementing a multi-label strategy where each patch could carry multiple labels. The assignment of these labels depends on whether the proportion of pixels that fall into a particular category exceeds a specified threshold.

## Methods

In this study, we propose an ensemble classifier to generate a multi-label hypothesis for every individual test input patch. The flowchart for the proposed ensemble classifier has been depicted in Fig 3. It consists of individual CNN-based one-vs-all classifiers to hypothesize the presence or absence of each corresponding class i.e. Gleason Grade 3, 4 or 5. The

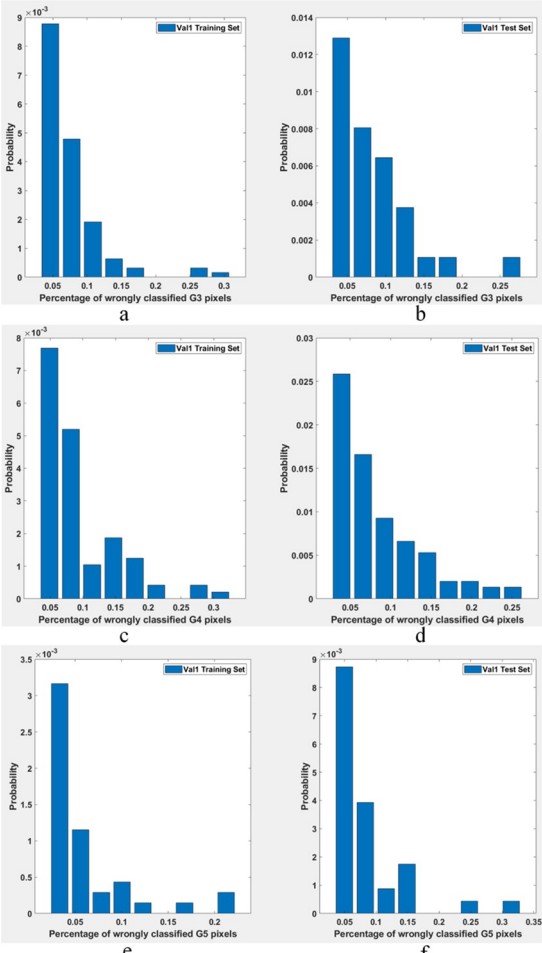

**Fig 2.** Probability distributions of misclassified pixel belonging to different classes in SICAPv2 dataset partition 'Val1' (a) G3 training; (b) G3 test; (c) G4 training; (d) G4 test; (e) G5 training; (f) G5 test.

rationalization behind the conception of this ensemble classifier is two-fold. First, each one-vs-all CNN classifier is deemed to perform better since all the layers (initial as well as final) will be devoted to feature extraction and classification specifically for each class. In contrast, a single multi-class CNN classifier shares the features extraction (initial layers) for all classes and only the head is devoted individually to each class. Second, multi-label approach to mitigate the labelling inconsistency problem can be efficiently handled by one-vs-all ensemble classifier especially since the number of classes are few and each classifier could be individually fine-tuned to address a single label.

The input to the ensemble classifier is a 512 × 512 image patch to be consistent with the default size of SICAPv2 dataset patches. Each individual classifier is a CNN dedicatedly trained for each of the three Gleason Grades i.e. G3, G4 and G5. These classifiers detect the corresponding class against the default Non-Cancerous (NC) class and all other grades (one-vs-all classifiers) if the individual detection score is above the selected threshold. The selection of the optimal thresholds has been described in the next section. Each classifier detects the presence (positive label) or absence (negative label) of its designated Gleason Grade independent of others. The multi-label output hypothesis is a concatenation of the respective outputs from each classifier. Thus, the multi-label output indicates presence or absence of any combination of

**Table 1. Summarized statistics related to misclassification of different classes in SICAPv2 dataset partitions.**

|  | Class | Average area of misclassified pixels | Maximum area of misclassified pixels |
|---|---|---|---|
| Val1 (Train) | G3 | 1.7% | 30% |
|  | G4 | 1.8% | 30% |
|  | G5 | 1.1% | 22% |
| Val1 (Test) | G3 | 1.6% | 27% |
|  | G4 | 1.8% | 25% |
|  | G5 | 1.8% | 30% |
| Val2 (Train) | G3 | 1.7% | 30% |
|  | G4 | 1.9% | 30% |
|  | G5 | 1.7% | 30% |
| Val2 (Test) | G3 | 1.3% | 25% |
|  | G4 | 1.4% | 25% |
|  | G5 | 0.3% | 7% |
| Val3 (Train) | G3 | 1.5% | 25% |
|  | G4 | 1.6% | 25% |
|  | G5 | 1.7% | 30% |
| Val3 (Test) | G3 | 1.7% | 30% |
|  | G4 | 1.9% | 30% |
|  | G5 | 1.2% | 22% |
| Test (Train) | G3 | 1.7% | 30% |
|  | G4 | 1.9% | 30% |
|  | G5 | 1.7% | 30% |
| Test (Test) | G3 | 1.1% | 18% |
|  | G4 | 1.0% | 17% |
|  | G5 | 0.5% | 10% |

G3, G4 and G5 Gleason Grades. This is in contrast to conventional multi-class classifiers which only output a single label even if multiple classes are present in the input patch. Due to the relatively small size of the dataset, a transfer learning approach is proposed to prevent over-fitting and training difficulties. For this purpose, different well-known CNN architectures such as ResNet18, ResNet50 and Inception etc. [34] pre-trained on ImageNet dataset [35] and their

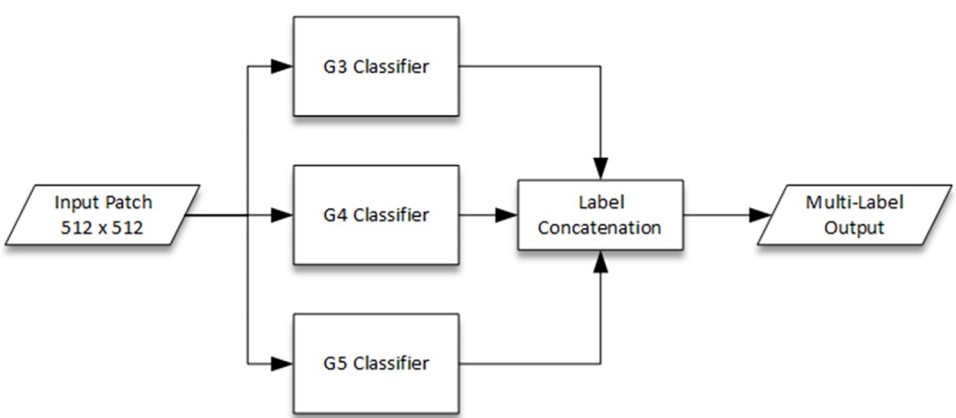

**Fig 3. Flowchart of the proposed ensemble classifier with individual CNN-based one-vs-all binary classifiers and multi-label output.**

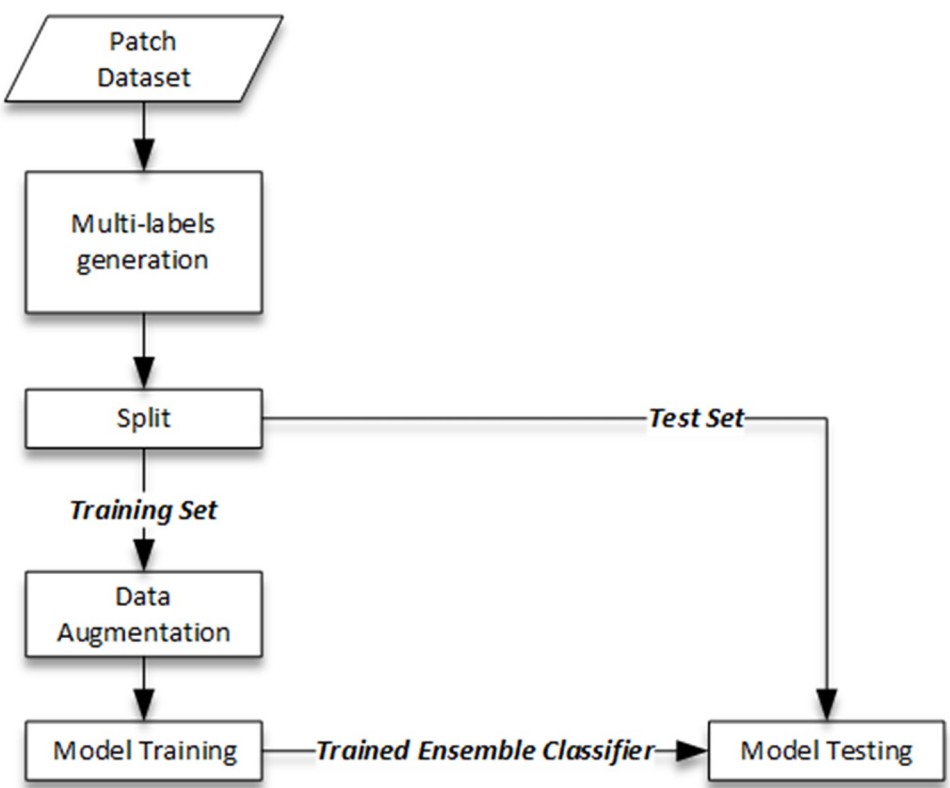

**Fig 4. Training and testing paradigm for the proposed multi-label ensemble classifiers.**

derivatives have been considered in the ablation study, presented in the next section, to select the best performing network.

The training and testing procedure of the proposed multi-label ensemble classifier has been depicted in Fig 4. Since the original dataset comes only with patch-level labels decided based on the majority votes, the first step is to generate multi-labels for each patch using the provided labelling masks. The label for each class (G3, G4 and G5) would be included in the output multi-label if the pixels corresponding to that class are above a certain percentage threshold. NC label is issued only if none of the pixels belonging to G3, G4 or G5 are present. Appropriate threshold values for training and testing respectively have been found using the ablation study described in the next section. The split between training and test sub-groups is based on the guidelines of the original dataset. The ensemble model is then trained by individually training the three component CNN one-vs-all models for each class i.e. G3, G4 and G5. The proposed multi-label ensemble classifier is then tested on the test examples for detection performance using standard metrics (e.g. accuracy and F1-Score etc.) as detailed in the next section.

A critical consideration while training the one-vs-all ensemble classifiers for multi-label scenario is the high data imbalance since the "all" category inevitably has many more training examples than the "one" category, leading to a bias towards the majority class. A potential solution to this problem is the use of a weighted cross-entropy loss function while training the models and has been adopted by Silva-Rodríguez et al. [29] even for their multi-class model since SICAPv2 is inherently an imbalanced dataset. Specifically, this function assigns more weight to under-represented classes and less weight to over-represented classes, penalizing the model more for misclassifying minority classes. However, setting these weights requires

careful consideration to avoid overfitting to the minority class. In this work, we have used ablation experiments to empirically determine the optimal weights. The findings of the ablation study have been presented in the next section.

The whole training and testing framework for the proposed ensemble classifier has been implemented in Matlab environment (R2022b) using Deep Learning Toolbox [34]. The computing environment is Intel(R) Core(TM) i9-9900K CPU @ 3.60GHz with 64 GB RAM and NVIDIA GeForce RTX 2080 Ti GPU.

## Results

This section presents the findings of our comprehensive study using the proposed ensemble classifier consisting of individual one-vs-all sub-classifiers for Gleason grade scoring task using multi-label approach. The following sub-section describes the ablation study conducted for a detailed analysis of the impact of various hyperparameters on the performance of the classifier. The patch-level Gleason grading results given later in this section demonstrate the effectiveness of our proposed method in classifying individual patches of histopathological images. Finally, the WSI-level labelling results illustrate the classifier's ability to accurately label entire histopathological slides. These results collectively highlight the robustness and efficacy of our proposed ensemble classifier in Gleason grade scoring.

### Ablation study

This sub-section presents the results of the ablation study conducted to tune various hyperparameters for the proposed ensemble classifier. These hyperparameters include the CNN architectures of the one-vs-all sub-classifiers, the threshold for assigning multi-labels at the patch level based on the pixel percentage belonging to a particular class, the number of epochs for training the model, the learning rate, and the L2 regularization factor. By systematically varying these hyperparameters, the influence of each on the model's performance has been studied and used to identify the final configuration suitable for the Gleason grade scoring task practically. For this purpose, the validation sets i.e. Val1, Val2, Val3 and Val4 of SICAPv2 datasets have been employed in all the experimentations.

While selecting the appropriate CNN architecture for each of the one-vs-all sub-classifiers to be used in the ensemble for each class, we have considered well-known CNNs e.g. ResNet18, ResNet50 and Inception pre-trained on ImageNet for transfer learning because these have learned robust feature representations, which can be leveraged to achieve high performance on our specific task with less data and training time. However, we observed a common trend of overfitting across these networks. Overfitting is a modeling error that occurs when a function is too closely fit to a limited set of data points and may therefore fail to predict additional data or future observations reliably. Fig 5 depicts one such scenario where we used ResNet18 as the sub-classifier for G3 grade on training set of 'Val1'. It can be seen that as the training progressed, the divergence between the training loss and validation loss increases which is a classic sign of overfitting. This problem can be mitigated through techniques such as using a lower complexity model, L2 regularization, early stopping, and data augmentation. To this end, we have employed standard image data augmentation techniques (resizing, rotation, translation and flipping) and early stopping if the validation loss increases for 8 consecutive epochs.

Additionally, given that even ResNet18 despite being the least complex among all considered CNNs led to overfitting, we have attempted to simplify it further. This was done by eliminating some of its final layers, thereby reducing its complexity and potentially making it less susceptible to overfitting. Specifically, the 'conv5' layer group and half of the 'conv4' layer group (i.e., '4b') of the standard ResNet18 architecture have been eliminated. This effectively reduced

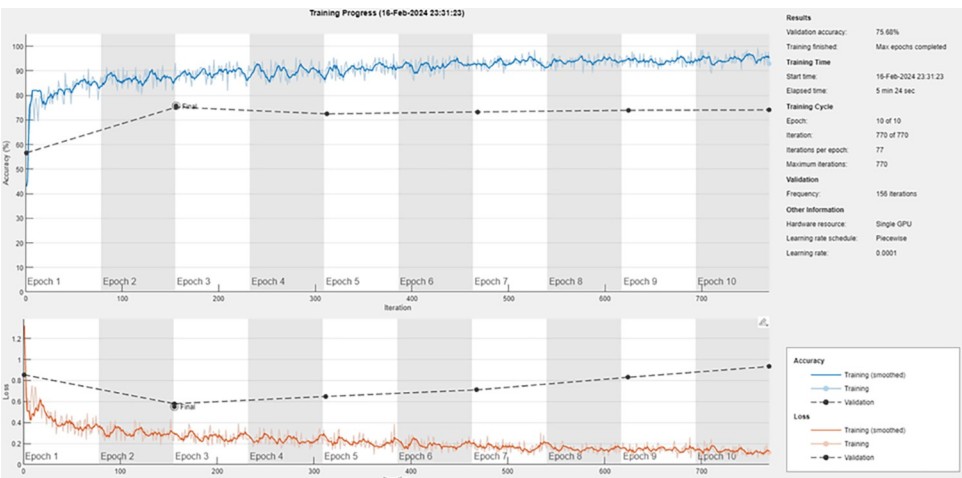

**Fig 5. Overfitting observed with ResNet18 CNN as sub-classifier for G3 grade classification in 'Val1' training set.**

the model's complexity, making it less prone to overfitting. Thus, the final Global Average Pooling (GAP) layer has been directly connected to the output of 'res4a_relu'. Finally, a fully connected and SoftMax layer for classification have been placed at the end of the proposed architecture. The architectural details of this proposed architecture have been given in Table 2. In addition to the aforementioned modifications, it's important to note that our initial attempts at mitigating overfitting by only removing the 'conv5'layer from ResNet18 were not successful. The model continued to overfit despite this simplification. On the other hand, when we further removed the 'res4a'layer, we observed a drop in the accuracy i.e. underfitting. Thus, the present architecture has been selected to strike a balance.

While training the proposed CNN architectures for each sub-class (G3, G4 and G5), a learning rate of '1e-3' was selected. This value was found to be optimal as higher learning rates led to suboptimal results, while lower rates required a greater number of epochs to converge. The learning rate has been scheduled to drop by 0.1 every 15th epoch. Moreover, the learning rate of the final fully connected layer is set to be '10' times higher than that of the initial layers, adhering to the practice of transfer learning. This approach ensures that the initial layers, which have been pre-trained on the ImageNet dataset, largely retain their learned weights. Meanwhile, the final layer can quickly adapt to the examples from the Sicapv2 dataset. Although convergence has been observed to be generally achieved after '15' epochs in all the conducted experiments, the training is extended to '50' epochs for extra measure. This additional training helps to fine-tune the model as the initial layers also gradually adopt to the

**Table 2. Architecture of the proposed CNN model for binary classification (one-vs-all).**

| Layer Name | Activation Size |
|---|---|
| Image Input | 224 × 224× 3 |
| Conv1 (7 × 7, 64) Stride 2, BN, Relu, MAP-2 | 112 × 112× 64 |
| Conv2a (3 × 3, 64) Stride 1, BN, Relu | 56 × 56× 64 |
| Conv2b (3 × 3, 64) Stride 1, BN, Relu | 56 × 56× 64 |
| Conv3a (3 × 3, 128) Stride 2, BN, Relu | 28 × 28× 128 |
| Conv3b (3 × 3, 128) Stride 1, BN, Relu | 28 × 28 × 128 |
| Conv4a (3 × 3, 256) Stride 1, Relu, GAP | 1 × 1 × 256 |
| Fully Connected, SoftMax | 2 |

**Table 3. Effect of L2 regularization parameter on F1-score (validation set).**

| Value | F1-Score |
|---|---|
| 1e-3 | 0.68 |
| 9e-3 | 0.69 |
| 9.5e-3 | 0.71 |
| 1e-2 | 0.71 |
| 1.2e-2 | 0.71 |
| 2e-2 | 0.68 |
| 1e-1 | 0.6 |

dataset and potentially improve its ability to generalize from the training data to unseen data. Adam optimizer has been used throughout all the experiments as the initial experiments with SGDM did not yield promising results.

Since, overfitting is a serious concern in Sicapv2 dataset owing to its smaller size, in order to identify the most effective L2 regularization parameter, the study involved a series of experiments on the four validation sets (Val1, Val2, Val3 and Val4), systematically varying the L2 regularization parameter to evaluate its impact on the model's performance. The results have been given in Table 3. For sake of brevity, only the experimental values around '1e-2' have been reported which was found to be the optimal value yielding the highest F1 score of '0.71'. Fig 6 depicts an example training loss curve after optimizing the hyperparameters mentioned above. It can be noticed that the overfitting has been managed effectively.

Finally, to assign multiple labels to each patch based on the percentage of pixels belonging to each class i.e. G3, G4 and G5, our experimental results have shown that the presence of even 1% pixels belonging to a particular class is enough for training and classification. For a patch size of $512 \times 512$, this corresponds to at least 2621 pixels. Using a higher threshold leads to elimination of too many training examples which leads to overfitting especially for G5 class which has too few example patches. On the other hand, a lower threshold means too few representative pixels in a given patch for extracting meaningful features.

The proposed ensemble classifier has been trained and tested on the SICAPv2 dataset using the obtained hyperparameters. The results have been reported in the next sub-section.

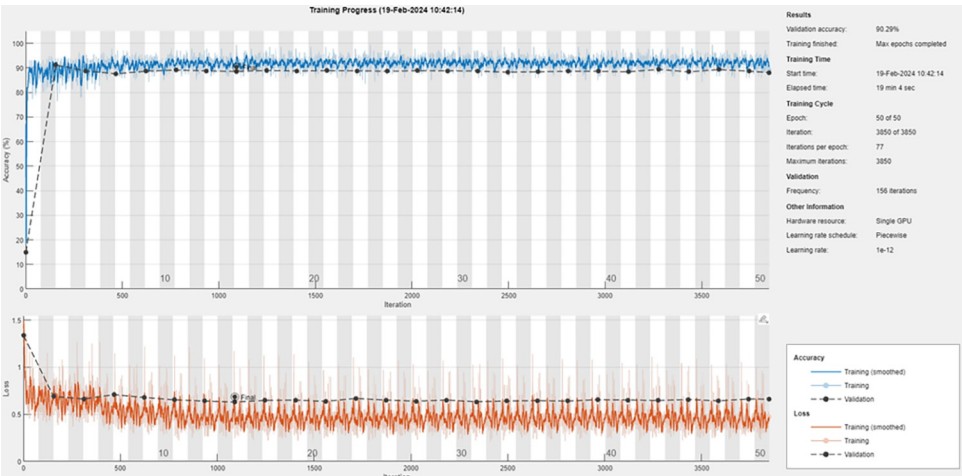

**Fig 6. Training loss curves for the proposed CNN architecture as sub-classifier for G3 grade classification in 'Val1' training set after hyperparameter optimization.**

**Table 4. Comparison of the proposed ensemble classifier on SICAPv2 dataset against reference works.**

| Model | Accuracy | F1-Score | | | | |
|---|---|---|---|---|---|---|
| | | Average | NC | G3 | G4 | G5 |
| **Test Set** | | | | | | |
| Proposed Ensemble Classifier | 0.85 | 0.71 | 0.85 | 0.69 | 0.75 | 0.54 |
| FSConv [29] | 0.67 | 0.65 | 0.86 | 0.59 | 0.54 | 0.61 |
| ProGleason-GAN [27] | 0.71 | 0.67 | - | - | - | - |
| WHOLESIGHT [26] | - | 0.66 | - | - | - | - |
| **Validation Set** | | | | | | |
| Proposed Ensemble | 0.87 | 0.75 | 0.83 | 0.72 | 0.77 | 0.69 |
| FSConv [29] | 0.76 | 0.71 | 0.88 | 0.73 | 0.71 | 0.54 |

## Patch-level Gleason grading results

The proposed ensemble classifier has been compared against the state-of-the-art works in Table 4. Every constituent one-vs-all sub-classifier is based on the CNN architecture depicted in Table 2 and has been trained four times for each of the validation and test sets of SICAPv2 datasets to ensure consistency of the results. Standard deviation of less than 0.05 has been observed in all the experiments which indicates high repeatability of the proposed approach. Accuracy and F1-scores have been reported in each case. Due to the imbalanced nature of the dataset, a higher F1-score is more important and indicative of a more robust model. It can be observed that the proposed model achieves higher accuracy as well as F1-score (average of all classes) than the recent works reported in the literature on both validation and test sets of SICAPv2 dataset.

Precision-recall curves for the individual sub-classifiers on validation and test sets have been plotted in Figs 7–11. F1-score has been overlayed as well. The values reported in Table 4 correspond to the best value obtained for each curve.

To gain further insight into the decision-making process of the trained CNN models, Figs 12–14 depict 'Grad-CAM' [36] visualization of the three sub-classifiers on three different example patches from SICAPv2 dataset respectively. Grad-CAM provides a visual explanation of the decision-making process of a CNN, which is crucial in medical imaging. Specifically, it generates a heatmap that highlights the significant regions in the input image that the CNN focuses on when making a prediction. This allows medical professionals to understand why a particular diagnosis was made. Fig 12 shows a patch (Example 1) containing only G5, labelled orange in the mask. The corresponding heat map for the G5 sub-classifier roughly corresponds to the labelled mask emphasizing the confidence in its utility.

The patch shown in Fig 13 (Example 2) contains both G4 and G5 categories (Cyan and Orange labels) and have been rightly classified by their corresponding sub-classifiers as indicated by their respective heatmaps. The heatmap for G5, however, significantly overlaps that of G4 indicating the similarities between these two classes. Fig 14 shows another interesting patch (Example 3) containing only G3 class (labelled yellow). The labelling area only makes up a small portion of the whole mask towards the bottom. Despite this, the corresponding heatmap for only G3 classifier shows a strong activation map.

Fig 15 shows the corresponding activation maps for the proposed classifier on a whole biopsy slide with Gleason grades labelled G4 and G5 as primary and secondary respectively.

## Discussion

The proposed multi-label classification approach for Gleason grading of prostate cancer at the patch-level by employing an ensemble of sub-classifiers to individually detect each of the three

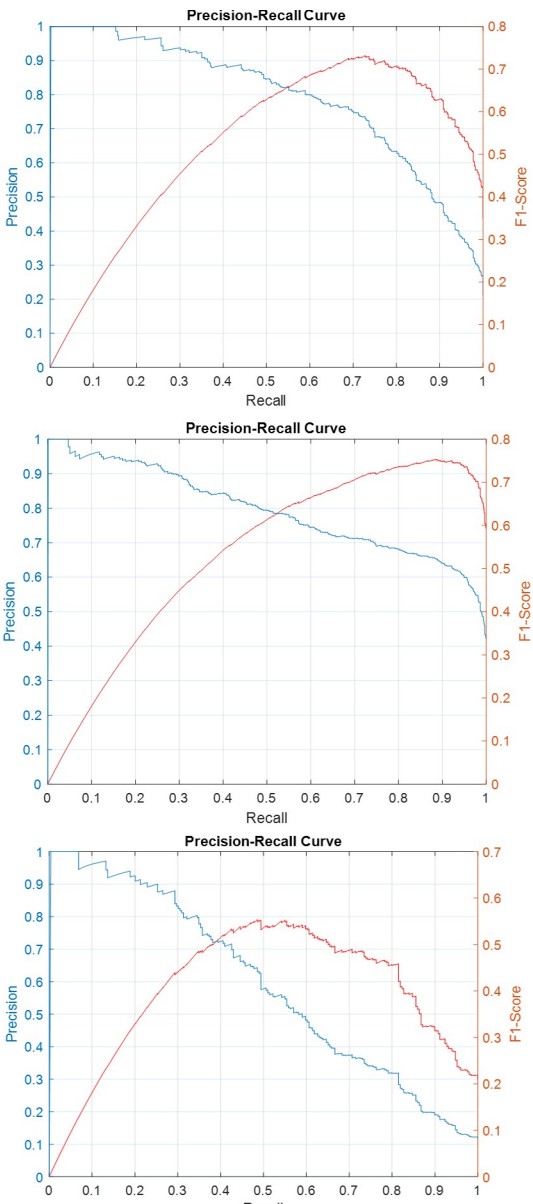

**Fig 7.** Precision-Recall and F1-Score curves for the sub-classifiers in the ensemble detector on SICAPv2 'test' set (a) G3 (b) G4 (c) G5.

Gleason grades (G3, G4, and G5) represents a departure from traditional multi-class classification techniques reported in the literature. Specifically, earlier reported works in the domain have commonly addressed the problem either by binary classification i.e. distinguishing between cancerous and non-cancerous slides, or as a multiclass problem involving the classification of slides at the patch level. We, on the other hand, have emphasized that at the patch level, labelling inconsistencies are inherent due to the majority voting and thus, the traditional classification approaches are not optimal. We have proposed the grading of prostate cancer histopathology slides as a multilabel classification problem since a patch may inevitably include pixels from multiple grades as show in **Fig 1**. The results presented in the previous section demonstrate the effectiveness of our proposed method since it achieved a significant

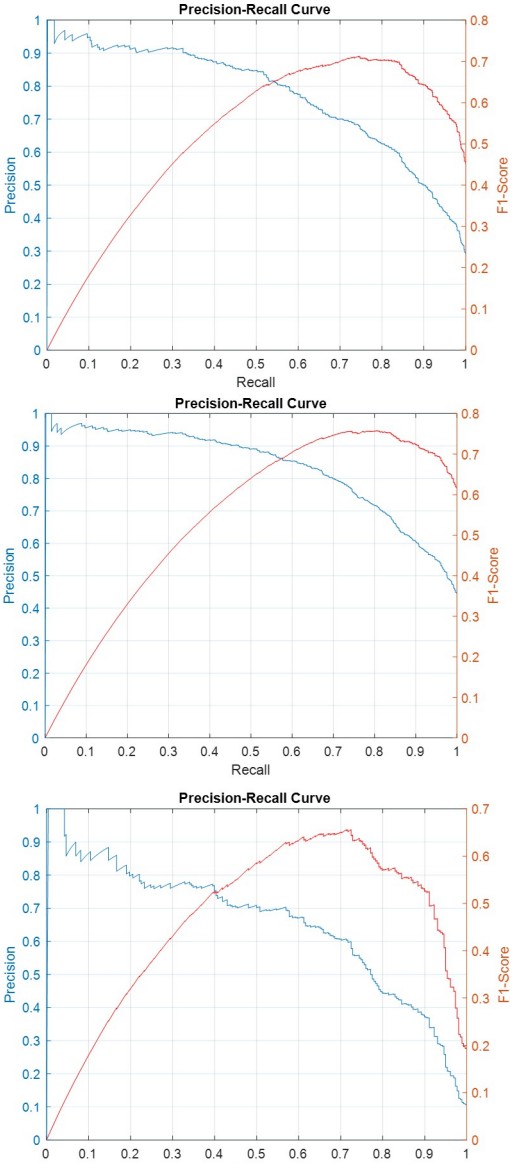

**Fig 8.** Precision-Recall and F1-Score curves for the sub-classifiers in the ensemble detector on SICAPv2 'val1' set (a) G3 (b) G4 (c) G5.

improvement over the state-of-the-art works on both test and validation sets of SICAPv2 dataset. Specifically, our model outperformed by achieving *14%* higher accuracy and a *4%* higher F1-score on the test set as highlighted by the data given in Table 4. Given the imbalance in the dataset, the F1-score becomes a more significant metric than accuracy. Our proposed classifier demonstrated superior performance on the G3 and G4 classes, achieving a higher F1-score compared to competing models. More importantly, despite the individual class performance, our model maintained a higher average F1-score. This indicates that our model is not only effective at identifying specific Gleason grades but also maintains a balanced performance across all classes, which is crucial in the context of imbalanced datasets. This further underscores the robustness and reliability of our proposed multi-label classification approach for Gleason grading in prostate cancer detection.

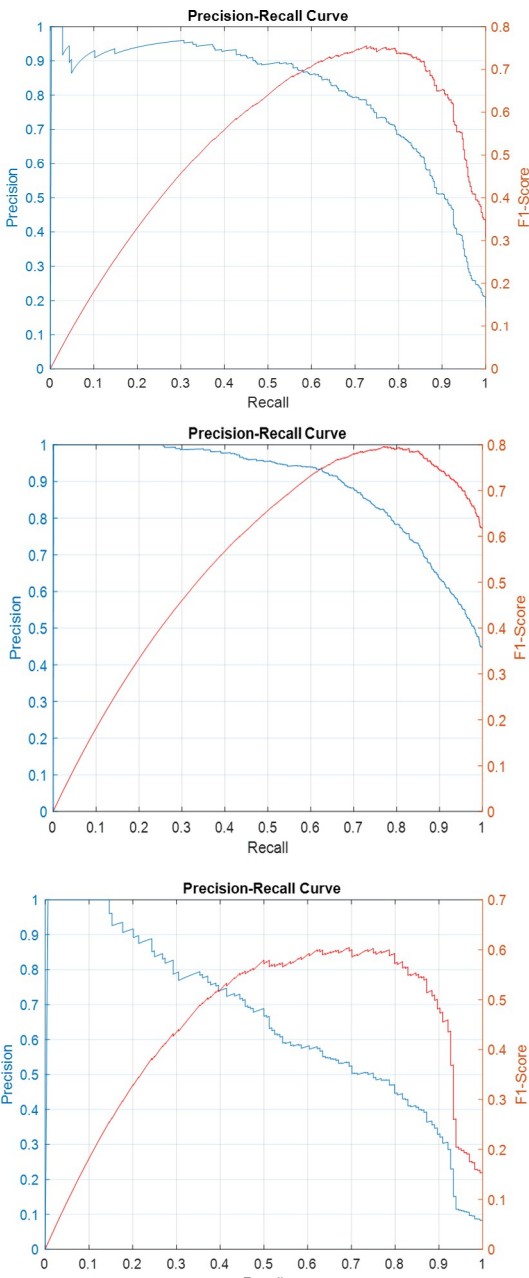

**Fig 9.** Precision-Recall and F1-Score curves for the sub-classifiers in the ensemble detector on SICAPv2 'val2' set (a) G3 (b) G4 (c) G5.

The lower F1-score for the G5 class on the test set can indeed be attributed to the significant class imbalance, with only 250 examples for G5 compared to 1873 for the remaining classes. This imbalance can skew the performance metrics and make it challenging to achieve high scores for underrepresented classes. However, it's encouraging to see that the performance on the validation set is better, with only the G3 classifier slightly underperforming i.e. *72%* compared to *73%* as reported in the literature for FSConv [29]. Moreover, despite these individual class performances, the average F1-score of our model is superior on both sets. This demonstrates the robustness of our proposed multi-label classification approach, even in the face of

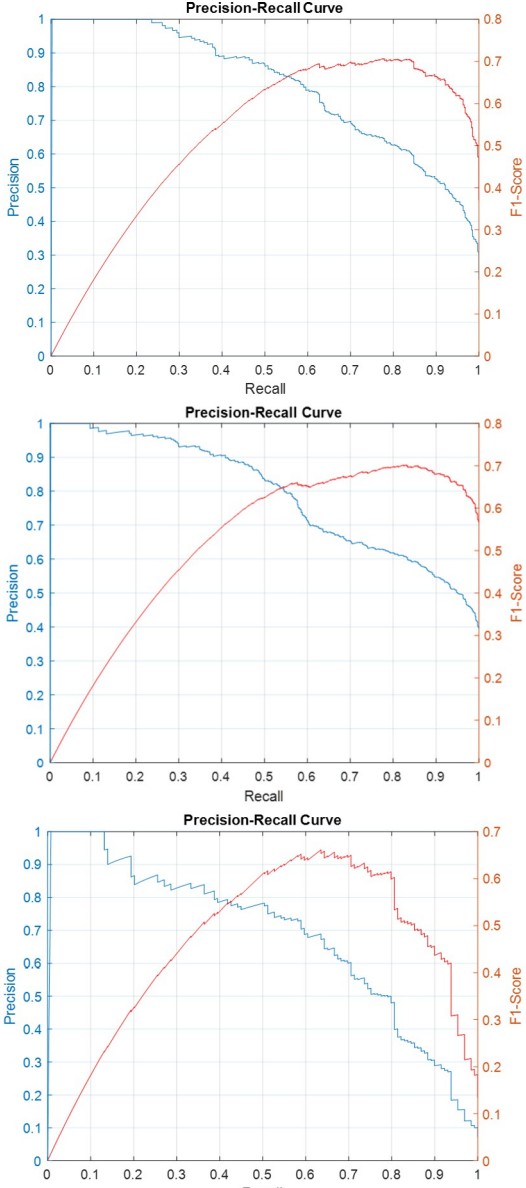

**Fig 10.** Precision-Recall and F1-Score curves for the sub-classifiers in the ensemble detector on SICAPv2 'val3' set (a) G3 (b) G4 (c) G5.

significant class imbalances. ProGleason-GAN [27] and WHOLESIGHT [26] have only provided results on the *Test* set of SICAPv2 dataset and it can be noticed from Table 4 that these are only marginally better on SICAPv2 dataset than the original work of FSConv [29] i.e. by *2%* and *1%* respectively in terms of average F1-score. Moreover, these works have not reported detailed scores for individual classes. These results clearly indicate the advantage of the proposed multi-label ensemble classifier over the ones reported in the literature.

Incorporating the important observation of pixels belonging to multiple classes being present in each patch, our study's results demonstrate the superiority of the multi-label approach over conventional multi-class classification for Gleason grade classification at the patch level. This is particularly evident in certain patches where pixels belonging to more than one class

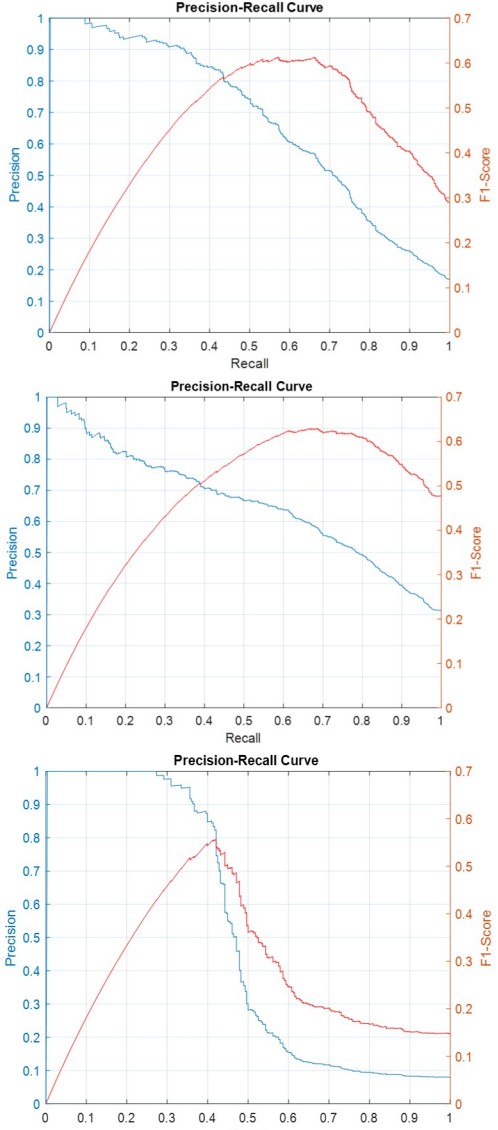

**Fig 11.** Precision-Recall and F1-Score curves for the sub-classifiers in the ensemble detector on SICAPv2 'val4' set (a) G3 (b) G4 (c) G5.

can be present, making the classification of a patch to just a single Gleason grade inappropriate. Despite the class imbalance, our multi-label approach achieved a higher average F1-score on both the test and validation sets, indicating effective identification of each Gleason grade independently. The multi-label approach proved more robust to class imbalance, achieving a higher average F1-score even with fewer examples of the G5 class. This robustness is crucial in medical imaging, where certain conditions may be underrepresented. The multi-label approach also allows for more fine-grained classification, treating each Gleason grade as a separate label, enabling more nuanced predictions beneficial at the patch level where subtle differences can be crucial for accurate diagnosis. The improved F1-scores for the G3 and G4 classes on the validation set further underscore the effectiveness of the multi-label approach. These results suggest that the multi-label approach provides a more accurate and robust method for

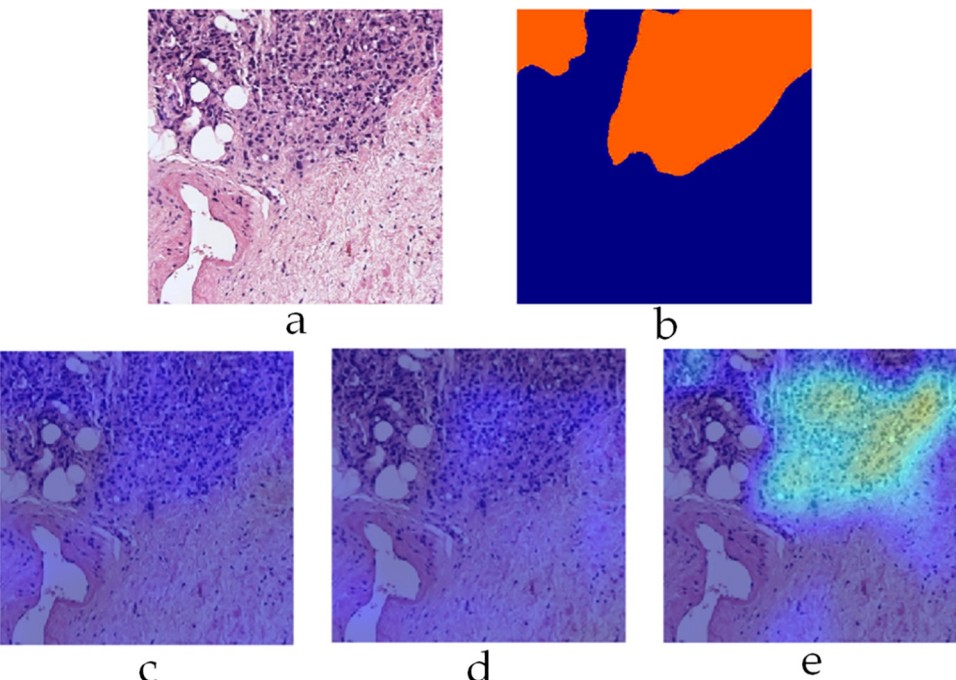

**Fig 12.** Grad-CAM visualization on example 1 a) input patch b) label mask c) G3 sub-classifier heat map d) G4 sub-classifier heat map e) G5 sub-classifier heat map.

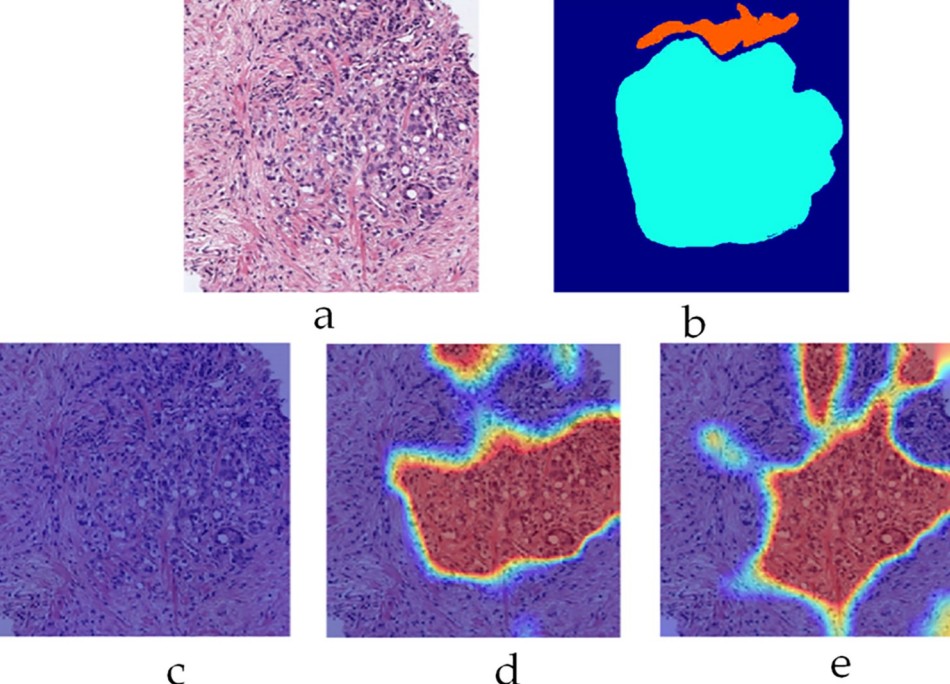

**Fig 13.** Grad-CAM visualization on example 2 a) input patch b) label mask c) G3 sub-classifier heat map d) G4 sub-classifier heat map e) G5 sub-classifier heat map.

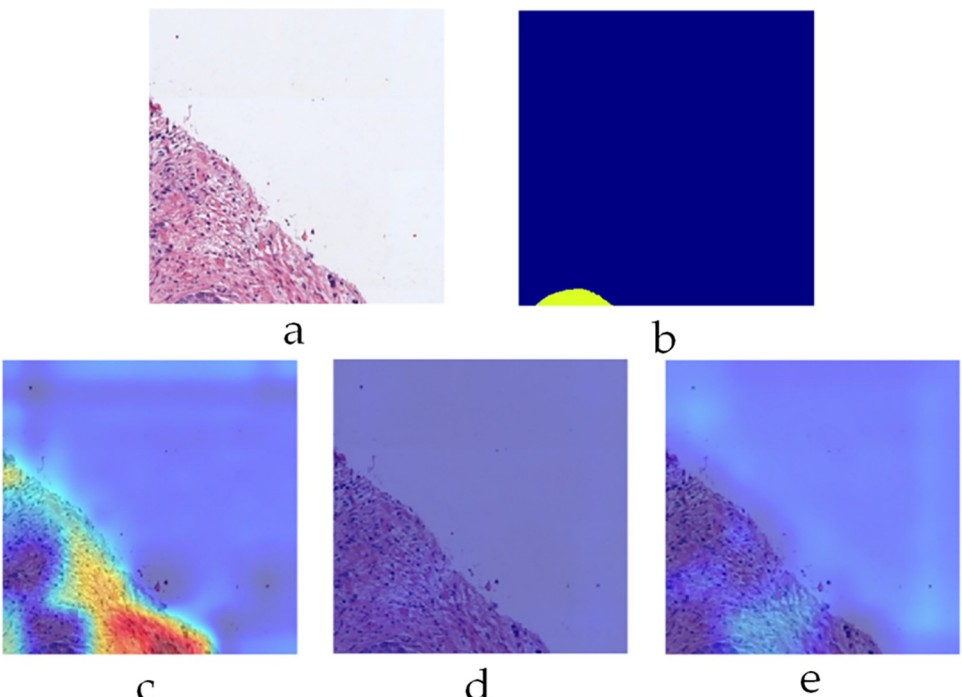

**Fig 14.** Grad-CAM visualization on example 3 a) input patch b) label mask c) G3 sub-classifier heat map d) G4 sub-classifier heat map e) G5 sub-classifier heat map.

Gleason grade classification at the patch level, making it a promising technique for improving the accuracy and reliability of prostate cancer detection.

## Conclusions

This study has proposed a multi-label ensemble deep-learning classifier to increase the accuracy of Gleason grading by effectively addressing the issue of label inconsistencies inherently present in the dataset patches. The proposed ensemble classifier consists of three one-vs-all sub-classifiers, fine-tuned variants of the ResNet18 CNN architecture, to accurately indicate the presence of one or more Gleason grades (G3, G4, and G5) in each patch. The experimental results demonstrate the superiority of our approach over traditional single-label classifiers, thereby enhancing the accuracy and consistency of Gleason grading. One potential improvement for the future tasks is deemed to be the segmentation of the labeling masks at pixel-level granularity, which could increase the accuracy of patch-level Gleason scoring. Additionally, the labeling noise due to manual annotation could be mitigated by generating labeling masks through the trained model and then re-verifying them through human experts. These enhancements could further improve the precision of Gleason grading and contribute to the ongoing efforts to leverage advanced machine learning techniques in cancer diagnostics. The proposed framework has been made available as open-source code to facilitate researchers and practitioners working in the field of digital histopathology.

In our future research, we plan to extend our experiments to include patch sizes other than the default 512 x 512 used in the original SICAPv2 dataset. We also aim to explore pixel-level segmentation alongside patch-based classification for a more detailed scoring of the entire slides. One limitation of the proposed ensemble classifier is the reliance on pre-trained networks to avoid overfitting through transfer learning. This is crucial given the

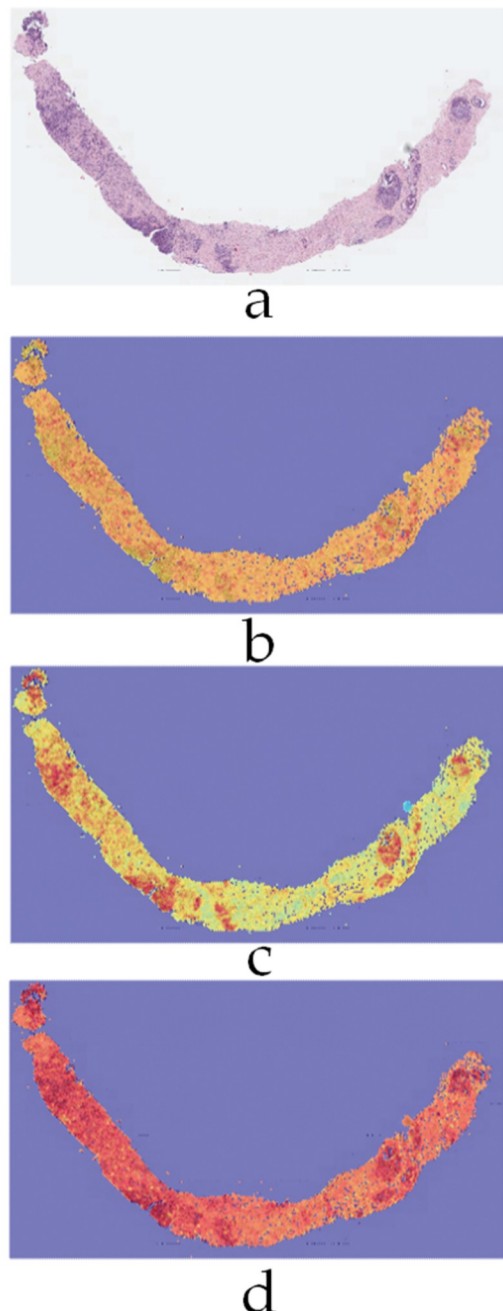

**Fig 15.** Activation maps on a WSI example a) Input image b) G3 c) G4 d) G5.

limited number of training examples in the SICAPv2 dataset. Although our experimental results have shown improvements over previous studies, we believe that further enhancements could be achieved by generating more synthetic examples using the generative models such as the one proposed by Golfe et al. [27, 28]. Consequently, having a larger number of training examples could allow us to train that are adapted to the problem at hand while training from scratch.

## Acknowledgments

The authors gratefully acknowledge technical and financial support provided by the Ministry of Education and Deanship of Scientific Research (DSR) at King Abdulaziz University, Jeddah, Saudi Arabia.

## Author Contributions

**Conceptualization:** Muhammad Asim Butt, Muhammad Farhat Kaleem, Muhammad Bilal.

**Data curation:** Muhammad Asim Butt, Muhammad Bilal.

**Formal analysis:** Muhammad Asim Butt, Muhammad Bilal.

**Funding acquisition:** Muhammad Bilal.

**Investigation:** Muhammad Asim Butt, Muhammad Farhat Kaleem, Muhammad Bilal.

**Methodology:** Muhammad Asim Butt, Muhammad Bilal.

**Project administration:** Muhammad Farhat Kaleem, Muhammad Bilal.

**Resources:** Muhammad Asim Butt, Muhammad Farhat Kaleem, Muhammad Shehzad Hanif.

**Software:** Muhammad Asim Butt.

**Supervision:** Muhammad Bilal.

**Validation:** Muhammad Asim Butt, Muhammad Shehzad Hanif.

**Visualization:** Muhammad Asim Butt.

**Writing – original draft:** Muhammad Asim Butt, Muhammad Farhat Kaleem, Muhammad Bilal.

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
