## [Decision Letter · Decision Letter 0]

3 Apr 2024

PONE-D-24-08473Using Multi-Label Ensemble CNN Classifiers to Mitigate Labelling Inconsistencies in Patch-level Gleason GradingPLOS ONE

Dear Dr. Bilal,

Thank you for submitting your manuscript to PLOS ONE. After careful consideration, we feel that it has merit but does not fully meet PLOS ONE’s publication criteria as it currently stands. Therefore, we invite you to submit a revised version of the manuscript that addresses the points raised during the review process.

Dear authors, some notices must be taken into consideration:

KEYWORDS must be alphabetic.The background in Section 2 must be combined with the introduction section.The "Discussion" should critically analyze the new findings in light of existing knowledge and available literature. Prepare this section carefully and make sure to highlight your contributions to the scientific literature.It is preferred to put a table in the discussion section to compare their work with other literature according to the algorithm, accuracy, and the used dataset.==============================

We look forward to receiving your revised manuscript.

Kind regards,

Hadeel K. Aljobouri

Academic Editor

PLOS ONE

Journal Requirements:

   "This research work was funded by Institutional Fund Projects under grant no. (IFPIP: 1825-135-1443). The au-thors gratefully acknowledge technical and financial support provided by the Ministry of Education and King Abdulaziz University, DSR, Jeddah, Saudi Arabia."

4. Please remove your figures from within your manuscript file, leaving only the individual TIFF/EPS image files, uploaded separately. These will be automatically included in the reviewers’ PDF.

Reviewers' comments:

Reviewer's Responses to Questions

**Comments to the Author**

1. Is the manuscript technically sound, and do the data support the conclusions?

Reviewer #1: No

Reviewer #2: Yes

2. Has the statistical analysis been performed appropriately and rigorously? 

Reviewer #1: No

Reviewer #2: N/A

3. Have the authors made all data underlying the findings in their manuscript fully available?

Reviewer #1: Yes

Reviewer #2: Yes

4. Is the manuscript presented in an intelligible fashion and written in standard English?

Reviewer #1: No

Reviewer #2: Yes

5. Review Comments to the Author

Reviewer #1: The content has not been found appropriate for a journal publication:

1. The motivation of the article is weakly focused.

2. The contributions of the article are not agreeable and, rather misleading.

3. Using a dataset is not your contribution.

4. The structure of the article is not appropriate, and results (preliminary tables and figures) are discussed before the methodology.

5. No novel approach was detected in the work.

6. The claims made about the technical content of ML/DL are disagreeable.

Reviewer #2: The abstract provides a clear overview of the problem addressed, the methodology employed, and the key findings, setting the stage for the reader's understanding and accurately reflects the focus of the research on addressing labeling inconsistencies in Gleason grading using multi-label ensemble CNN classifiers.

The introduction was clear and relevant to the context of the research problem and the author provided a sufficient background on the significance of the problem. The background includes key studies and methodologies in the effectiveness of patch-level Gleason Grading of prostate histopathology images demonstrating a deep understanding of the existing literature.

The methods section has a detailed description of the deep learning model architecture and training process. Although, some information in the results is recommended to be added to the methods instead and present the validated data in the results section. Also, the descriptive flowchart may include more information on the stated method to give a clearer image of the proposed process. Some figures require double-checking of the captions and resolution to improve presented information.

The data used in the paper is explained and referenced to be available in future work as well as the evaluation metrics, and validation procedures. The results presentation was clear through tables, figures, and performance metrics to reflect the effectiveness of the multi-label ensemble CNN classifiers. An expansion of limitations and future directions to guide further research in the field is recommended.

The references support the background, methodology, and findings of the research with double-checking the accuracy of references, including author names, publication years, journal/conference titles, and page numbers, to ensure credibility and readability are recommended.

Moreover, comments are posted on the reviwed article provided in the attached file.

6. PLOS authors have the option to publish the peer review history of their article (what does this mean?). If published, this will include your full peer review and any attached files.

Reviewer #1: **Yes: **Prof Dr Shahzad Ahmad Qureshi

Reviewer #2: No

---

## [Author Response · Author response to Decision Letter 0]

18 May 2024

Academic Editor

Dear authors, some notices must be taken into consideration:

Comment #1: “KEYWORDS must be alphabetic.”

Author’s Response: Keywords have been listed in alphabetical order in the revised manuscript. 

Comment #2: “The background in Section 2 must be combined with the introduction section.”

Author’s Response: The literature review (Section 2 – Background in the previously submitted version of the manuscript) has been merged with the introduction section as advised. 

Comment #3: “The "Discussion" should critically analyze the new findings in light of existing knowledge and available literature. Prepare this section carefully and make sure to highlight your contributions to the scientific literature.”

Author’s Response: The section “Discussion” has been enhanced to emphasize the contribution of the presented work in the context of the earlier works published in the literature. Specifically, it has been highlighted that 

“the earlier reported works in the domain have commonly addressed the problem either by binary classification i.e. distinguishing between cancerous and non-cancerous slides, or as a multiclass problem involving the classification of slides at the patch level. We, on the other hand, have emphasized that at the patch level, labelling in-consistencies are inherent due to the majority voting and thus, the traditional classification approaches are not optimal. We have proposed the grading of prostate cancer histopathology slides as a multilabel classification problem since a patch may inevitably include pixels from multiple grades as show in Fig 1. The results presented in the previous section demonstrate the effectiveness of our proposed method since it achieved a significant improvement over the state-of-the-art works on both test and validation sets of SICAPv2 dataset. Specifically, our model outperformed by achieving 14% higher accuracy and a 4% higher F1-score on the test set as highlighted by the data given in Table 4.”

Moreover, the conclusion section also discusses how the proposed approach is robust against the inherent class imbalance problem as depicted through discussion of concrete F1-score values.

Comment #4: “It is preferred to put a table in the discussion section to compare their work with other literature according to the algorithm, accuracy, and the used dataset.”

Author’s Response: The authors appreciate the significance of the provided suggestion to highlight the utility of the proposed ensemble classifier. Table 4 in the subsection “Patch-Level Gleason Grading Results” of the main section “Results” compares the performance of the proposed algorithm on SICAPv2 dataset. In the revised manuscript, the comparison results provided in this table have been further discussed in section “Discussion”. Specifically, it has been mentioned that:

“The lower F1-score for the G5 class on the test set can indeed be attributed to the significant class imbalance, with only 250 examples for G5 compared to 1873 for the remaining classes. This imbalance can skew the performance metrics and make it challenging to achieve high scores for underrepresented classes. However, it's encouraging to see that the performance on the validation set is better, with only the G3 classifier slightly underperforming i.e. 72% compared to 73% as reported in the literature for FSConv [29]. However, despite these individual class performances, the average F1-score of our model is superior on both sets. This demonstrates the robustness of our proposed multi-label classification approach, even in the face of significant class imbalances. ProGleason-GAN [27] and WHOLESIGHT [26] have only provided results on the Test set of SICAPv2 dataset and it can be noticed from Table 4 that these are only marginally better on SICAPv2 dataset than the original work of FSConv [29] i.e. by 2% and 1% respectively in terms of average F1-score. Moreover, these works have not reported detailed scores for individual classes. These results clearly indicate the advantage of the proposed multi-label ensemble classifier over the ones reported in the literature.”

Journal Requirements:

Comment #1. Please ensure that your manuscript meets PLOS ONE's style requirements, including those for file naming. The PLOS ONE style templates can be found at 

Author’s Response: We have made amendments to the manuscript to conform to the author affiliations, headings, tables and figure captions etc. according to the information given in the above online documents. 

Comment #2. Please note that funding information should not appear in any section or other areas of your manuscript. We will only publish funding information present in the Funding Statement section of the online submission form. Please remove any funding-related text from the manuscript.

Author’s Response: The funding agency requires us to put the following statement on the manuscript without any changes. This is the same statement that has been put in the “Financial Disclosure” section of the online submission form. If it appears as it is anywhere on the manuscript, then it satisfies the funding agency requirement. 

“This research work was funded by Institutional Fund Projects under grant no. (IFPIP: 1825-135-1443). The authors gratefully acknowledge technical and financial support provided by the Ministry of Education and King Abdulaziz University, DSR, Jeddah, Saudi Arabia.”

Comment #3. Thank you for stating the following financial disclosure: 

 "This research work was funded by Institutional Fund Projects under grant no. (IFPIP: 1825-135-1443). The authors gratefully acknowledge technical and financial support provided by the Ministry of Education and King Abdulaziz University, DSR, Jeddah, Saudi Arabia."

Author’s Response: Please include the following statement as “Role of Funder” in the online submission form. 

Comment #4. Please remove your figures from within your manuscript file, leaving only the individual TIFF/EPS image files, uploaded separately. These will be automatically included in the reviewers’ PDF.

Author’s Response: We have removed the figures from the main file in the revised manuscript.

Reviewer 1

Comment #1: “The motivation of the article is weakly focused.”

Author’s Response: Thank you for your efforts in reviewing the manuscript and highlighting the shortcomings. The main motivation of the presented study is the problem of labelling inconsistency in the patch-based Gleason grading due to majority voting-based label assignment as considered by the works reported in the literature. To this end, multi-label ensemble classifier has been proposed which has been shown to give better classification accuracy as demonstrated through the provided results. The revised manuscript highlights the motivation of the proposed work towards the end of the introduction as follows:

“Specifically, the patch-level labelling has been conventionally done through majority vote of how each pixel in the patch is labeled according to the Gleason grade. This approach, however, leads to at least three problems i.e.

Loss of Information: The majority voting scheme potentially ignores the information related to the minority classes which are inevitably present in numerous patches

Misclassification: If the patch contains a mix of different Gleason grades, the majority voting scheme could result in misclassification since it is highly sensitive to the manual pixel-level labelling done by the pathologists

Labelling Noise: The presence of label noise can negatively affect the training performance of the machine learning models which rely heavily on the accuracy of the provided labels at the patch level”

Comment #2-3: “The contributions of the article are not agreeable and, rather misleading.”

“Using a dataset is not your contribution.”

Author’s Response: To elaborate this point further, the revised manuscript lists the main contributions of the paper as follows:

“• Provides a statistical analysis of the patch-level labelling noise in SICAPv2 prostate histology dataset.

• Proposes an ensemble machine learning classifier to detect all occurrences (multi-labels) of Gleason grades at the patch level, rather than just the majority grade.

• Provides an open-source framework for Gleason grading at patch and WSI-level based on the pro-posed ensemble classifier to facilitate researchers and practitioners working in the field of digital histopathology.”

To this end, figure 2 and Table 1 in the paper provide insight into the labelling inconsistencies of the prostate WSI SICAPv2 dataset. Specifically, 

“It can be noticed that while on average only up to 2% of pixels are misclassified in a given set, as many as 30% could be misclassified in individual instances. This misclassification can potentially lead to suboptimal performance while training a machine learning classifier especially in the wake of high level of imbalance.”

Furthermore, the results provided in Table 2 clearly show that the proposed ensemble classifier based on multi-labelling at the patch level achieves state of the art results on SICAPv2 dataset. The resubmitted manuscript has been revised to emphasize the contributions in the “Discussion” section. 

Comment #4: “The structure of the article is not appropriate, and results (preliminary tables and figures) are discussed before the methodology.”

Author’s Response: Thanks for the suggestions to improve the tutorial content of the manuscript. Based on the suggestions from the reviewers and the associate editor, the literature review section has been merged with the introduction. The contributions of the work have been listed at the end of the introduction section. Moreover, the conclusion section has been enhanced in the revised manuscript to further discuss the results. The proposed approach i.e. multi-label ensemble classifier derives its motivation from the statistical insights into the patch-level labelling inconsistencies present in the dataset. Thus, it seems prudent to demonstrate these statistical figures to illustrate the level of labelling noise (Fig. 1, Fig. 2 and Table 1) before the introduction of the working of the proposed classification strategy. The results related to the empirical determination of the appropriate thresholds to assign the multi-labels have been provided in the ablation study later. To elaborate this point further, the sub-section has been renamed as “Statistical Insight into Patch-level Labelling Inconsistencies” and the following text appears at the end:

“In the light of these observations, this study suggests implementing a multi-label strategy where each patch could carry multiple labels. The assignment of these labels depends on whether the proportion of pixels that fall into a particular category exceeds a specified threshold.”

Comment #5: “No novel approach was detected in the work.”

Author’s Response: The main contribution of the proposed work is the introduction of “multi-label classification” at the patch level which is important due to the inherent labelling inconsistencies in the dataset. This aspect has been highlighted in the “Discussion” section of the revised manuscript as follows:

“The proposed multi-label classification approach for Gleason grading of prostate cancer at the patch-level by employing an ensemble of sub-classifiers to individually detect each of the three Gleason grades (G3, G4, and G5) represents a departure from traditional multi-class classification techniques reported in the literature. Specifically, earlier reported works in the domain have commonly addressed the problem either by binary classification i.e. distinguishing between cancerous and non-cancerous slides, or as a multiclass problem in-volving the classification of slides at the patch level. We, on the other hand, have emphasized that at the patch level, labelling inconsistencies are inherent due to the majority voting and thus, the traditional classification approaches are not optimal. We have proposed the grading of prostate cancer histopathology slides as a mul-tilabel classification problem since a patch may inevitably include pixels from multiple grades as show in Fig 1. The results presented in the previous section demonstrate the effectiveness of our proposed method since it achieved a significant improvement over the state-of-the-art works on both test and validation sets of SI-CAPv2 dataset. Specifically, our model outperformed by achieving 14% higher accuracy and a 4% higher F1-score on the test set as highlighted by the data given in Table 4.”

Comment #6: “The claims made about the technical content of ML/DL are disagreeable.”

Author’s Response: The main premise of the proposed approach for patch-level Gleason Grade classification is the underlying labelling noise as noted in the manuscript as:

“SICAPv2 dataset has provided patch-level labels to facilitate classification. This scheme requires that if a patch contains more than one annotated grade, the label typically assigned is the majority grade.”

This observation has been made by earlier researchers as well. For instance, the manuscript notes that:

“Recently, Morales-Álvarez et al. [20] have proposed the use of multiple instance learning to tackle the problem of patch-level label generation by exploiting the correlation among neighboring patches. Although 95.11% accuracy has been reported on SICAPv2 dataset, this study is limited to the binary cancer detection task and Gleason grading has not been considered.”

To elaborate the point further, we have added further relevant references and emphasized the contribution of the proposed work in the revised manuscript as follows:

“The labelling noise in histopathology datasets has been recognized by researchers and various techniques have been suggested to mitigate the effects of the same. For instance, Karimi et al. [32] have suggested model regularization to alleviate this problem. Similarly, Ashraf et al. [33] have proposed a unique loss-based method for de-noising patch-based labels for Gastric cancer histopathology images. However, the problem of labelling noise due to the presence of more than one type of Gleason Grades has not been addressed.”

To highlight the significance of the presented work, the Discussion section of the revised manuscript notes that:

“The proposed multi-label classification approach for Gleason grading of prostate cancer at the patch-level by employing an ensemble of sub-classifiers to individually detect each of the three Gleason grades (G3, G4, and G5) represents a departure from traditional multi-class classification techniques reported in the literature. Specifically, earlier reported works in the domain have commonly addressed the problem either by binary classification i.e. distinguishing between cancerous and non-cancerous slides, or as a multiclass problem in-volving the classification of slides at the patch level. We, on the other hand, have emphasized that at the patch level, labelling inconsistencies are inherent due to the majority voting and thus, the traditional classification approaches are not optimal. We have proposed the grading of prostate cancer histopathology slides as a mul-tilabel classification problem since a patch may inevitably include pixels from multiple grades as show in Fig 1. The results presented in the previous section demonstrate the effectiveness of our proposed method since it achieved a significant improvement over the state-of-the-art works on both test and validati

---

## [Editor Report · Decision Letter 1]

21 May 2024

Using Multi-Label Ensemble CNN Classifiers to Mitigate Labelling Inconsistencies in Patch-level Gleason Grading

PONE-D-24-08473R1

Dear Dr. Bilal,

We’re pleased to inform you that your manuscript has been judged scientifically suitable for publication and will be formally accepted for publication once it meets all outstanding technical requirements.

Kind regards,

Hadeel K. Aljobouri

Academic Editor

PLOS ONE
---

## [Editor Report · Acceptance letter]

30 May 2024

PONE-D-24-08473R1 

PLOS ONE

Dear Dr. Bilal, 

I'm pleased to inform you that your manuscript has been deemed suitable for publication in PLOS ONE. Congratulations! Your manuscript is now being handed over to our production team.

Kind regards, 

on behalf of

Asst.Prof.Dr. Hadeel K. Aljobouri 

Academic Editor

PLOS ONE